

# Larval exposure to field-realistic concentrations of clothianidin has no effect on development rate, over-winter survival or adult metabolic rate in a solitary bee, *Osmia bicornis*

Elizabeth Nicholls[1], Robert Fowler[1], Jeremy E. Niven[1], James D. Gilbert[1,2] and Dave Goulson[1]

[1] School of Life Sciences, University of Sussex, Falmer, Brighton, United Kingdom
[2] School of Biological, Biomedical and Environmental Sciences, University of Hull, Hull, United Kingdom

## ABSTRACT

There is widespread concern regarding the effects of agro-chemical exposure on bee health, of which neonicotinoids, systemic insecticides detected in the pollen and nectar of both crops and wildflowers, have been the most strongly debated. The majority of studies examining the effect of neonicotinoids on bees have focussed on social species, namely honey bees and bumble bees. However, most bee species are solitary, their life histories differing considerably from these social species, and thus it is possible that their susceptibility to pesticides may be quite different. Studies that have included solitary bees have produced mixed results regarding the impact of neonicotinoid exposure on survival and reproductive success. While the majority of studies have focused on the effects of adult exposure, bees are also likely to be exposed as larvae via the consumption of contaminated pollen. Here we examined the effect of exposure of *Osmia bicornis* larvae to a range of field-realistic concentrations (0–10 ppb) of the neonicotinoid clothianidin, observing no effect on larval development time, overwintering survival or adult weight. Flow-through respirometry was used to test for latent effects of larval exposure on adult physiological function. We observed differences between male and female bees in the propensity to engage in discontinuous gas exchange; however, no effect of larval clothianidin exposure was observed. Our results suggest that previously reported adverse effects of neonicotinoids on *O. bicornis* are most likely mediated by impacts on adults.

# INTRODUCTION

Bees are important pollinators of crops and wild flowers, therefore ongoing population declines and extinctions are a major cause for concern, particularly considering increasing global reliance on insect-pollinated crops (*Holden, 2006*; *Gross, 2008*; *Aizen & Harder, 2009*). Such declines are likely attributable to a number of factors, including habitat loss,

Corresponding author
Elizabeth Nicholls,
E.Nicholls@sussex.ac.uk

the spread of non-native species, emergent pathogens and perhaps most controversially, exposure to pesticides (*Goulson, Lye & Darvill, 2008*; *Goulson et al., 2015*; *Ollerton et al., 2014*; *Vanbergen, 2013*). Of the multitude of chemicals applied to arable land, neonicotinoids are some of the most widely used insecticides (*Jeschke et al., 2011*; *Simon-Delso et al., 2014*) and the class most strongly implicated in bee declines (*Sanchez-Bayo & Goka, 2014*; *Woodcock et al., 2016*). Typically applied as a seed treatment to oilseed and cereal crops, these systemic insecticides become incorporated into all tissues of the plant as it grows, including pollen and nectar, providing a direct route of oral exposure for bees and other pollinators.

Neonicotinoids are neuro-active insecticides which target nicotinic acetylcholine (nACh) receptors in the insect nervous system, causing over-stimulation of nerve cells, paralysis and at sufficiently high doses, death (*Tomizawa & Casida, 2005*; *Palmer et al., 2013*; *Moffat et al., 2015*). Laboratory studies have found the oral toxicity of neonicotinoids to be relatively high for bees (4–5 ng/honeybee, *Suchail, Guez & Belzunces, 2001*), and sub-lethal effects have also been observed following exposure to concentrations within the range detected in field-collected pollen and nectar (reviewed in *Alkassab & Kirchner, 2017*). These effects include deficits in learning (*Williamson & Wright, 2013*; *Stanley et al., 2015*), foraging (*Feltham, Park & Goulson, 2014*; *Gill & Raine, 2014*) and homing ability (*Henry et al., 2012*), all of which are essential to bee survival and reproduction.

While previous research has typically focussed on the effects of adult exposure to acute doses via contaminated nectar, the frequent detection of neonicotinoid residues in the pollen of both crops and wildflowers suggests that bees are likely to be exposed to neonicotinoids throughout their entire life cycle (*Mullin et al., 2010*; *Dively et al., 2015*; *Botías et al., 2015*; *David et al., 2016*; *Ellis et al., 2017*), and recent studies have shown that the severity of the effect can be dependent on both the timing and duration of exposure (*Suchail, Guez & Belzunces, 2001*; *Decourtye, Lacassie & Pham-Delègue, 2003*; *Rortais et al., 2005*; *Rondeau et al., 2014*; *Heard et al., 2017*). Though underinvestigated at present, studies have shown that early exposure to neonicotinoids can negatively impact on larval development (*Wu, Anelli & Sheppard, 2011*; *Gregorc et al., 2012*; *Derecka et al., 2013*; *Rondeau et al., 2014*; *Rosa et al., 2016*) and may also have latent effects on adult physiology and behaviour (*Yang et al., 2012*; *Tomé et al., 2012*; *Tan et al., 2015*; *Peng & Yang, 2016*).

The limited body of research investigating the effects of chronic or developmental exposure to neonicotinoids may arise in part from a bias towards studying bees that are commercially reared for pollination, namely honey bees and bumble bees (*Lundin et al., 2015*), which are all social species. While the biology of these species is well understood, it can be challenging to study the effects of larval exposure to pesticides independently of worker effects, or to monitor individuals over their entire lifetime. Assessing effects on reproduction in these social species is also complicated given that the unit of replication is the colony. Moreover, it remains unclear whether these managed species actually serve as a good proxy for the diversity of bee species that likely come into contact with pesticides when foraging and nesting in arable landscapes (*Devillers et al., 2003*; *Scott-dupree, Conroy & Harris, 2009*; *Biddinger et al., 2013*; *Rondeau et al., 2014*; *Arena & Sgolastra, 2014*; *Heard et al., 2017*). The majority of wild bees are solitary and, consequently, have life histories
that differ considerably from those of managed pollinators. Flight periods and foraging preferences are highly variable between species and so the degree of exposure to pesticides is also likely to differ. Add to this inherent differences in physiology among species, including between honey bees and bumble bees (*Cresswell et al., 2012*; *Piiroinen & Goulson, 2016*), and it becomes clear that much more research is needed to determine the sensitivity of wild bees to neonicotinoid exposure, as well as the implications of both larval and chronic exposure on bee development, adult functioning and ultimately, the stability of bee populations.

Here we investigate the effects of chronic, developmental exposure to a neonicotinoid in a solitary, cavity nesting bee, *Osmia bicornis*. A pollen generalist, the flight period of *O. bicornis* overlaps considerably with that of winter-sown oilseed rape, and studies have shown that *O. bicornis* can greatly benefit from this mass-flowering crop in terms of reproductive output (*Jauker et al., 2012*; *Holzschuh et al., 2013*; *Diekötter et al., 2014*). However, concerns have been raised regarding the trade-off between increased food availability and exposure to pesticides such as neonicotinoids (*Rundlöf et al., 2015*). The small number of studies of neonicotinoid exposure in this species have so far yielded mixed results regarding their sensitivity. A large-scale field experiment by *Rundlöf et al. (2015)* found a severe effect of proximity to clothianidin-treated spring-sown oilseed rape on *O.bicornis* nesting success, as well as wild bee density more generally. A laboratory-based study by *Sandrock et al. (2014)* found that adult *O. bicornis* exposed to nectar containing a combination of thiamethoxam (2.87 ppb) and clothianidin (0.45 ppb) also had reduced reproductive success. In contrast, *Peters, Gao & Zumkier (2016)* found little effect of proximity to winter-sown, clothanidin-treated oilseed rape on *O. bicornis* reproduction. Levels of clothianidin in pollen sampled from nests close to treated oilseed rape were found to be low (1–1.7 ppb), with the majority of pollen samples containing residues below the limit of quantification (LOQ), though it should be noted that pollen was sampled at a single time point during the flowering season (23 days after cocoon placement and the start of oilseed rape full flowering) and pooled across nests, therefore individual larva developing in nest cells provisioned earlier or later in the season may have experienced differing levels of exposure. None of the aforementioned studies directly examined the effect of neonicotinoid exposure on *O. bicornis* larval development.

Female *O. bicornis,* like many other species of cavity nesting bee, provision their offspring with a single mass of unprocessed pollen, providing an opportunity to manipulate and tightly control pesticide exposure throughout development without the confound of adult or worker exposure (*Abbott et al., 2008*; *Konrad et al., 2008*; *Hodgson, Pitts-Singer & Barbour, 2011*). In this study, we spiked pollen with a range of clothianidin concentrations (0–10 ppb) representative of the levels detected in the pollen of both oilseed rape (Mean = 2.27 ppb, Range =≤ 0.12–14.5 ppb, *Botías et al., 2015*) and field margin plants likely to be visited by *O. bicornis* (Range = 1.1–9.4 ppb, *Krupke et al., 2012*; Mean = 1.2 ppb, Range = 0–5.9 ppb, *Rundlöf et al., 2015*; Range ≤ 0.12–0.36 ppb, *Botías et al., 2015*) and monitored effects on larval development time, survival and overwintering success. Because larval exposure to neuro-toxic compounds may affect nervous system development and basic autonomic functioning (*Dwyer, McQuown & Leslie, 2009*; *Peng & Yang, 2016*), we

also tested for latent effects of larval exposure on adult physiology. Cellular metabolism which underpins all physiological processes, is reliant on the delivery of oxygen to tissues and the removal of carbon dioxide, which in insects occurs via the tracheal system. The rate of gas exchange is mediated by the opening and closing of the spiracles, which is under neuronal control. Many insects exhibit cyclical or discontinuous patterns of gas exchange, where spiracles are kept closed for prolonged periods, hypothesised to be a strategy to minimise water loss when resting (*Buck, Keister & Specht, 1953*; *Buck & Keister, 1955*; *Quinlan & Hadley, 1993*; *Quinlan & Lighton, 1999*). Across a diversity of insects, exposure to neuro-active pesticides has been show to affect rates and patterns of gas exchange, such as the propensity to engage in discontinuous gas exchange (reviewed in *Karise & Mänd, 2015*). In bees, acute exposure to imidacloprid has been shown to alter abdominal ventilation patterns in adult honey bees (*Hatjina et al., 2013*) and to increase metabolic rates in the stingless bee *Melipona quadrifasciata* (*Tomé et al., 2015*). Accordingly, patterns of respiratory gas exchange are considered a useful physiological measure of an insects' response to stress, therefore we used flow-through respirometry to examine the metabolic rate and respirometry rhythms of adult *O. bicornis* exposed to clothianidin as larvae.

## METHODS

### Study organism

*Osmia bicornis* (Linneaus, 1758) is a solitary bee that nests in dead plant stems. Adults emerge in early spring, whenever temperatures exceed *ca.* 12 °C. Following mating, females begin provisioning nests with pollen and nectar (hereafter referred to as the pollen provision), and once a sufficiently large provision of pollen has been collected, an egg is laid and then a mud partition is built to form an individual nest cell (*Raw, 1972*). The female then begins provisioning another cell, and so on, until the tube is full, at which point the female seals the entrance with mud. Within a nest tube, female eggs tend to be laid first, and provisioned with more pollen, with male eggs and their smaller provisions found towards the entrance of the nest. Approximately one week after laying, eggs hatch and the larvae begin to eat the pollen provision. Once all the pollen is consumed (after *ca.* 30 days) the larvae spin a cocoon and pupate, overwintering as an adult inside the cocoon and emerging the following spring (*Raw, 1972*).

### Rearing methods

Six 'trap nests', consisting of cardboard tubes (8 mm diameter, 150 mm in length) housed in a waterproof shelter, were positioned a few metres above ground in an orchard at Stanmer Organics, East Sussex, UK. Stanmer Organics has been Soil Association certified organic for the past 10 years. Nests were placed out in early April 2016, and each contained a release tube seeded with 12 female and 10 male cocoons. Cocoons were checked for viability prior to release by making a small incision at the tip of the cocoon, and clypeal hair colour was used to distinguish between male and female bees. From mid-May onwards, once bees had emerged and females had begun provisioning the nests, tubes were checked daily for the presence of eggs. Eggs plus pollen provisions were removed from the nests and added to individual polystyrene nest blocks, which had Perspex lids to permit observation of larval

development. A small piece of cotton wool was used to plug the entrance to the nests, and nests were covered when in the field, to limit light exposure.

Nest blocks were then returned to the laboratory and pollen provision (plus egg) were weighed to the nearest 0.001 g (Precisa 125A; Newport Pagnell, Bucks, UK) before being placed into a dark incubator (20 °C, 75% RH). Bees were assigned to one of four treatments; 0 ppb (control), 1 ppb, 3 ppb or 10 ppb clothianidin, with care taken to balance the provision weight and nest position (as a proxy for sex) across treatments.

## Pesticide exposure

Clothianidin is currently the most commonly used seed treatment worldwide, and is a breakdown product of another commonly applied neonicotinoid, thiamethoxam (*Simon-Delso et al., 2014*). Data on neonicotinoid residue levels commonly present in the nests of *O. bicornis* was unfortunately lacking at the time of the experiment (but see *Peters, Gao & Zumkier, 2016* which detected concentrations of 1–1.7 ppb clothianidin in *O. bicornis* pollen provisions) so we selected a range of clothianidin doses (0–10 ppb) to reflect the range of concentrations commonly detected in field-collected oilseed rape and wildflower pollen (*Botías et al., 2015*; *Rundlöf et al., 2015*; *Krupke et al., 2012*), with 10 ppb serving as a 'worst-case scenario' level of exposure, though still within the range of concentrations detected in the pollen of crops and wildflowers. To contaminate pollen provisions, 10 mg of technical grade clothianidin (Sigma-Aldrich, Gillingham, UK) was diluted in 10 ml of acetone to give an initial 1 mg/ml stock, which was then further diluted with acetone to produce a stock of 0.01 mg/ml, both of which were stored at −80 °C. On the day of pollen collection, 0.01 mg/ml stock was diluted with a mixture of acetone and water (10% acetone v/v) to give a 100 ng/ml solution, which was used to contaminate provisions in the 10 ppb treatment. This stock was further diluted to 30 ng/ml and 10 ng/ml to contaminate pollen in the 3 and 1 ppb treatments, respectively. Approximately 50 µL of solution was injected into a longitudinal fissure made in each pollen provision, with the exact volume varied according to provision weight to standardise the concentration within treatments. Pollen provisions in the 0 ppb group were injected with acetone and water alone. To test the accuracy of the spiking method and degradation of the compounds over time, a sub-sample of pollen from each treatment was frozen for residue analysis at −80 °C, either after 24 h or 28 days of incubation. Pollen samples were extracted using the QuEChERS method and analysed for neonicotinoid residues using ultra high-performance liquid chromatography tandem mass spectrometry (UHPLC-MS/MS) as described in *Botías et al. (2015)*. Samples were screened not only for clothianidin, but also for four other commonly applied neonicotinoids: thiamethoxam, imidacloprid, acetamiprid and thiacloprid (see Table S1 for method detection limits, quantification limits and absolute recoveries).

## Monitoring development

Following contamination of pollen provisions, nest blocks were returned to the incubator and checked daily for the following developmental stages: egg hatching, defecation, total pollen consumption, initiation of cocoon spinning, and cocoon completion. Larvae were reweighed once all pollen had been consumed, and the efficiency of food conversion was

calculated as the difference between the body weight of the mature larvae and the fresh pollen provision. Cocoons were weighed ten days after completion (once fully darkened) to give an estimate of pre-pupal weight. One hundred and twenty days after the beginning of the neonicotinoid exposure, the temperature in the incubator was reduced to 14 °C. Three weeks later, all nest blocks were moved to a cool climate controlled chamber held at 4 °C and 50% relative humidity (RH). Following a 196 day overwintering period, bees inside their cocoons were placed back into an incubator, warmed to 20 °C and checked daily for emergence.

## Metabolic rate measurement

Metabolic rate was measured for each bee on the day of emergence using flow-through respirometry, using $CO_2$ production as the measure of metabolic rate. Atmospheric air was scrubbed of $CO_2$ and $H_2O$ using a Drierite®-Ascarite® column, before being pumped through two chambers of identical volume (2 ml), one of which was used for measuring $CO_2$ production, the other serving as a reference. Flow rate through the chambers was maintained at 200 ml/minute via two mass flow controllers (GFC17; Aalborg, NY, USA). After passing through the chambers, air flowed into two separate channels of an infrared $CO_2$–$H_2O$ analyser (Li-7000; Li-Cor, Lincoln, NE, USA), and the output signal from the two analysers was captured by LiCor software. The sampling rate was 5 Hz. The temperature in the room was held at 21 °C (Mean $\pm$ $SD = 20.82 \pm 2.11$ ° C) and measurements lasted for 30 min per bee. This included an initial period of $CO_2$ stabilisation after opening and closing the chamber, and time for bees to recover from being transferred. Once the bee was secured inside the chamber the experimenter left the room to minimise disturbance. We observed no changes in bee activity whilst in the chambers during pilot studies (data not shown). Indeed, bees remained stationary throughout the recording period. Following metabolic rate measurements, bees were removed from the chamber and immediately weighed to the nearest 0.001 g (Precisa 125A, Newport Pagnell, Bucks, UK).

Respirometry data was analysed using OriginPro 2016 software (Origin Lab, Northampton, MA, USA). Volumes of $CO_2$ (ppm) were baseline corrected and temperature normalised using the Q10 correction for temperature differences (*Morgan, Shelly & Kimsey, 1985*). To calculate the rate of $CO_2$ production per bee, volume $CO_2$ (ppm) was converted to $CO_2$ fraction and then multiplied by the flow rate (200 ml min$^{-1}$). The integral of $CO_2$ min$^{-1}$ *vs.* min was calculated for a stable period of the recording, totalling approximately 20 min per bee. This value was then divided by the exact measurement time for each bee (*ca.* 0.33 h) to give a rate of ml $CO_2$ h $^{-1}$. Patterns of $CO_2$ production over time were visually inspected for each bee and categorised as either continuous, discontinuous or cyclic, though no observations of cyclic gas exchange were observed (*Buck, 1958*; *Lighton, 1996*; *Chown et al., 2006*; *Kovac et al., 2007*).

## Statistical analysis

All statistical analyses were conducted using SPSS v.22. Normality was assessed via inspection of q–q plots combined with the Shapiro–Wilk statistic. Homogeneity of variance was assessed via Levene's test. Where necessary data were transformed to meet

**Table 1  Concentration of clothianidin detected in pollen provisions, 24 h and 28 days after spiking provisions.** Provisions were incubated under the same conditions as bees ($n = 3$ per treatment per time point).

| CLO (ppb) | Time since application | |
|---|---|---|
| | 24 Hours (Mean ppb ± SE) | 28 Days (Mean ppb ± SE) |
| 0 | <MDL | <MDL |
| 1 | 0.741 ± 0.11 | 0.774 ± 0.11 |
| 3 | 2.567 ± 0.21 | 2.240 ± 0.26 |
| 10 | 8.958 ± 0.07 | 8.765 ± 0.33 |

Notes.
MDL, Method Detection Limit; SE, Standard Error.

the assumptions of parametric tests, or non-parametric alternatives were used. Fisher's exact test was used to compare mortality (number of dead *vs.* live bees) across clothianidin treatments during larval, pupal and overwintering stages of development. Pollen provision weight was log-transformed and compared across treatments using analysis of variance (ANOVA). There was no correlation between pollen provision weight and the time to cocoon spinning, so ANOVA was used to compare larval development time. Time to adult emergence was analysed using a two-way ANOVA with treatment and sex as the predictors. Larval and adult body mass was compared between treatments using analysis of co-variance (ANCOVA), with log-transformed pollen provision weight included as the co-variate. Given that provision weight is highly dependent on sex, the analysis was run separately for male and female bees. Homogeneity of regression slopes was assessed by testing for an interaction between provision weight and the categorical predictor, treatment (in all cases $p > 0.05$). Weight loss over winter was compared using a mixed-design ANOVA, with body mass (pre-pupa *vs.* adult) as the within-subjects measure and treatment and sex as the between-subjects predictors. Metabolic rate data was log-transformed and analysed using an ANCOVA, with neonicotinoid treatment and gas exchange pattern (discontinuous *vs.* continuous) as the predictors and log-transformed adult body mass as the co-variate. Again, because adult body mass differs between the sexes, the analysis was run separately for male and female bees.

## RESULTS

### Residue analysis

Control (0 ppb) pollen provisions contained no neonicotinoid residues, and no neonicotinoids other than clothianidin were detected in samples from each of the treatment groups, confirming that pollen collected by females at the field site was free of neonicotinoids (Table 1). Comparison of pollen provisions that were frozen for residue analysis after either 24 h or 28 days of incubation showed little evidence of degradation during this time, meaning bees were exposed to neonicotinoids throughout larval development (Wilcoxon Signed-Rank test, 24 h *vs.* 28 days; 1 ppb, $Z = 0.000, p = 1.000$; 3 ppb, $Z = -0.535, p = 0.593$; 10 ppb, $Z = -0.535, p = 0.593$).

**Table 2 Development time and larval, pupal and overwinter survival rates of *Osmia bicornis* reared on pollen provisions spiked with different concentrations of clothianidin.**

| CLO (ppb) | N Eggs | Development time (Days) | | % Mortality (Larval) | N Cocoons | % Mortality (Pupal) | % Mortality (Overwinter) | N Emerged adults | | % Survival (Overall) |
|---|---|---|---|---|---|---|---|---|---|---|
| | | Mean ± SD | Range | | | | | m | f | |
| 0 | 31 | 31 ± 5 | 24–41 | 6.45 | 29 | 13.79 | 8.80 | 8 | 14 | 70.97 |
| 1 | 33 | 31 ± 4 | 23–39 | 9.09 | 30 | 6.67 | 7.14 | 14 | 12 | 78.78 |
| 3 | 38 | 30 ± 4 | 24–42 | 9.21 | 35 | 5.71 | 6.06 | 15 | 16 | 81.58 |
| 10 | 36 | 32 ± 4 | 21–38 | 8.88 | 32 | 6.25 | 0.00 | 14 | 16 | 83.33 |

## Mortality

Of the 161 eggs collected, 25 were excluded from analysis either due to mechanical damage ($n = 9$), mould ($n = 7$) or the presence of parasites ($n = 9$). At each developmental stage, a maximum of four bees died per treatment (Table 2) and there was no significant difference in mortality between treatment groups during larval development ($\chi^2 = 0.837, df = 3, p = 0.882$), pupation ($\chi^2 = 4.351, df = 3, p = 0.214$) adult overwintering ($\chi^2 = 2.212, df = 3, p = 0.605$), or across all stages of development combined ($\chi^2 = 1.718, df = 3, p = 0.652$).

## Development time

Larval development time, defined as the number of days from egg hatching to the initiation of cocoon spinning, did not differ between treatments (Table 2; ANOVA, $F_{3,115} = 0.526$, $p = 0.665$). Bees took between 21 and 42 days to develop, with a mean of 30–32 days across all treatment groups (Mean ± SD; Control = 31 ± 5; 1 ppb = 31 ± 4; 3 ppb = 30 ± 4; 10 ppb = 32 ± 4).

## Larval food conversion and over-winter weight loss

Initial pollen provision weight did not differ between treatments (Table S2, ANOVA, $F_{3,133} = 0.053, p = 0.984$). As might be expected, provision weight had a strong effect on resulting larval body mass (ANCOVA, Pollen Weight (co-variate), $F_{1,116} = 258.941, p > 0.001$), but there was no difference in larval mass between clothianidin treatments (Fig. 1; ANCOVA, Treatment, $F_{3,101} = 0.722, p = 0.541$) suggesting that exposure had no effect on the ability of bees to convert pollen into wet body mass. Overall, female larvae had a higher body mass than males (Mann–Whitney, $U = 145, Z = -8.329, p < 0.001$; Median ± SD Female = 334.50 ± 57.24 mg; Male = 225.8 ± 30.12 mg),

Following overwintering, cocoons were placed back into an incubator at 20 °C and checked daily for emergence. As in the wild, males emerged first (ANOVA, Sex, $F_{1,103} = 169.099, p < 0.001$), between 0–14 days after warming (Mean ± SD = 6 ± 4 days), and females emerged after 6–26 days (Mean ± SD = 16 ± 3 days). There was no effect of clothianidin treatment on emergence time (Table S3; ANOVA, Treatment, $F_{3,103} = 0.095, p = 0.963$).

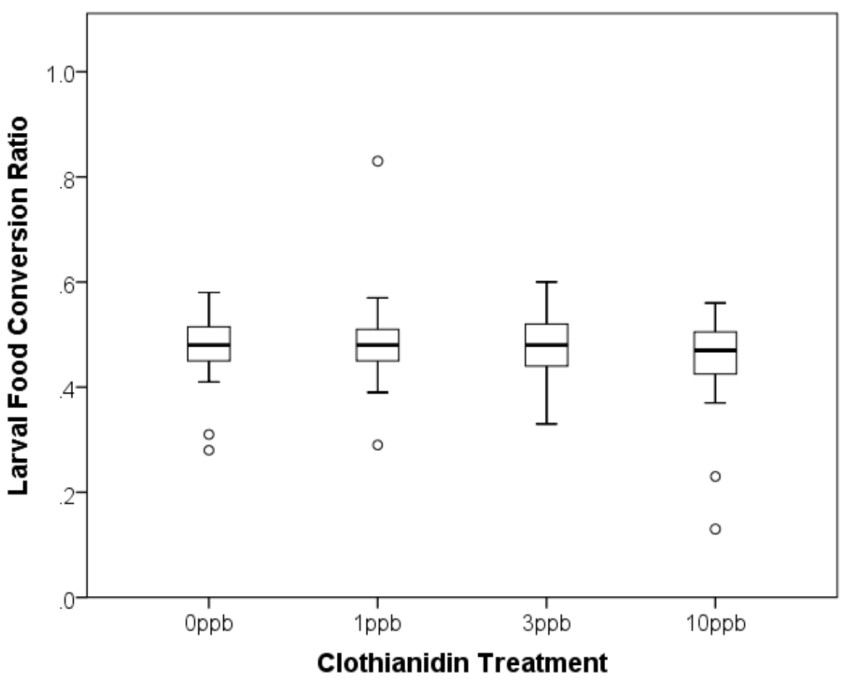

**Figure 1** **Efficiency with which bees converted pollen provision mass into larval body mass (median and interquartile range).** Pollen provisions were spiked with clothianidin at one of four doses (0 ppb $n = $ 29; 1 ppb $n = 30$; 3 ppb $n = 35$; 10 ppb $n = 32$). There was no difference in larval body mass between treatments.

Adult body mass did not differ between treatments (Fig. 2A; ANCOVA, Treatment, Female $F_{3,50} = 1.830, p = 0.154$; Males, $F_{3,49} = 0.769, p = 0.517$), nor did the amount of weight lost over winter (Fig. 2B; Mixed ANOVA, Time × Treatment, $F_{3,95} = 0.750, p = 0.525$). In absolute terms, female bees lost more weight than males (Mixed ANOVA, Time × Sex, $F_{1,95} = 34.637, p < 0.001$; Mean ± SD, Females = 47.64 ± 11.38 mg, Males = 36.80 ± 6.49 mg), but as a percentage of pupal weight, weight loss was similar between the sexes (Mean ± SD, Females = 36.04 ± 5.54% ; Males = 39.90 ± 4.98%).

## Metabolic rates

Previous studies have shown that pesticide exposure can alter gas exchange patterns in insects as well as the propensity to engage in discontinuous gas exchange (see *Karise & Mänd, 2015* for review). Therefore, the proportion of bees engaging in continuous (CGE, Fig. 3A) *versus* discontinuous (DGE, Fig. 3B) gas exchange was compared across clothianidin treatments using a 3-way loglinear analysis (Factors: Breathing Pattern, Sex, Treatment). This produced a final model which retained only the Breathing Pattern × Sex interaction (Likelihood ratio $\chi^2 = 6.686, df = 12, p = 0.878$), indicating that there was no significant difference between treatments in the propensity to engage in CGE but that overall, female bees were 2.7 times more likely to perform CGE than males (Table S4, Breathing Pattern × Sex, $\chi^2 = 4.149, df = 1, p = 0.042$).

There was no significant interaction between breathing pattern and body mass for either sex, indicating a similar mass-scaling of metabolic rate between bees engaged in CGE

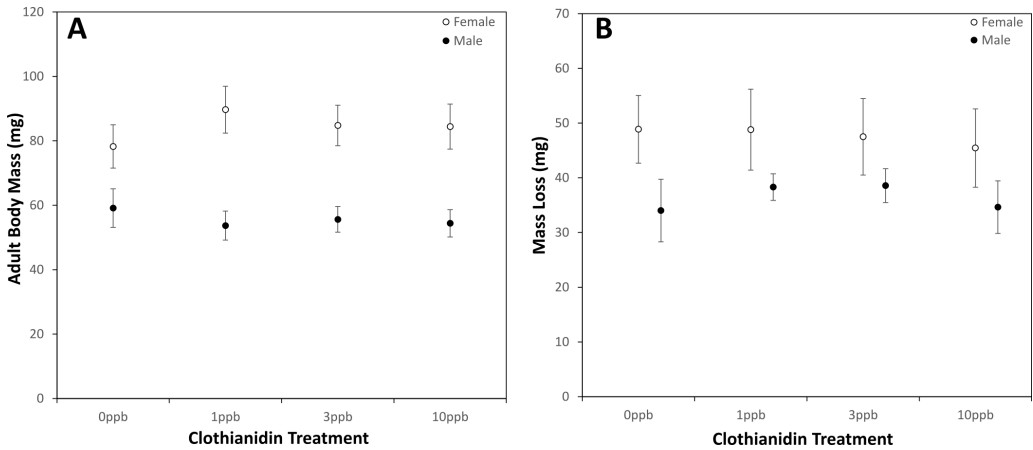

**Figure 2** **(A) Emergent adult body mass and (B) absolute mass loss over winter (pre-pupa to adult) for female (open circles, $n = 58$) and male (black circles, $n = 51$) bees exposed to pollen spiked with clothianidin during larval development at one of four doses (Mean ± 95% confidence interval (CI); 0 ppb $m = 8, f = 14$; 1 ppb $m = 14, f = 12$; 3 ppb $m = 15, f = 16$; 10 ppb $m = 14, f = 16$).** Data in (A) are least-square means estimated by an ANCOVA performed for each sex, with treatment as a main effect and log-transformed pollen mass as a co-variate (Females provision weight = 325.69 mg, Males = 222.23 mg).

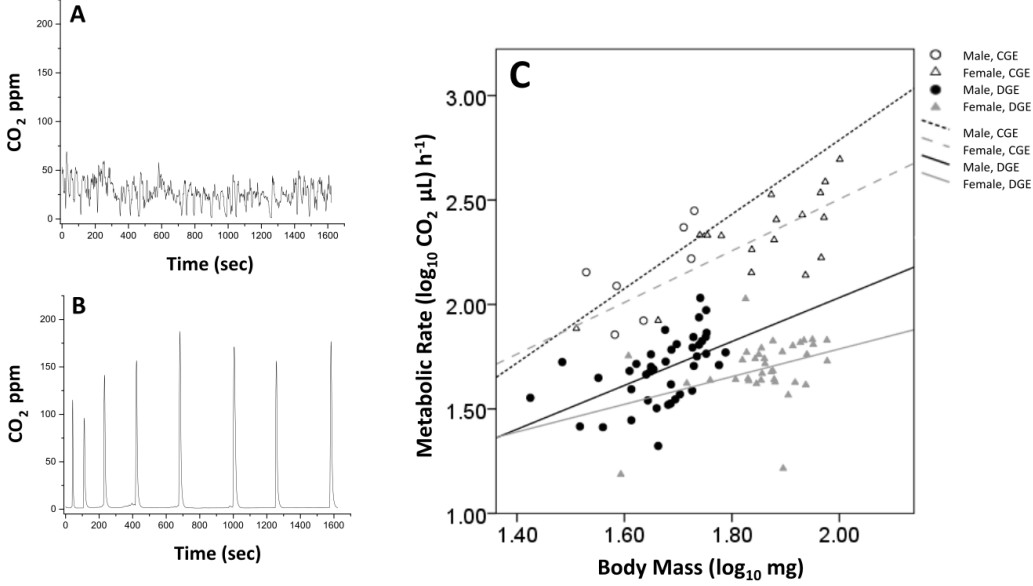

**Figure 3** **Example of continuous (A) and discontinuous patterns of gas exchange (B). Scaling relationship between metabolic rate and body mass for male (circles) and female (triangles) bees engaged in continuous (CGE, open shapes) *versus* discontinuous (DGE, filled shapes) gas exchange (C).**

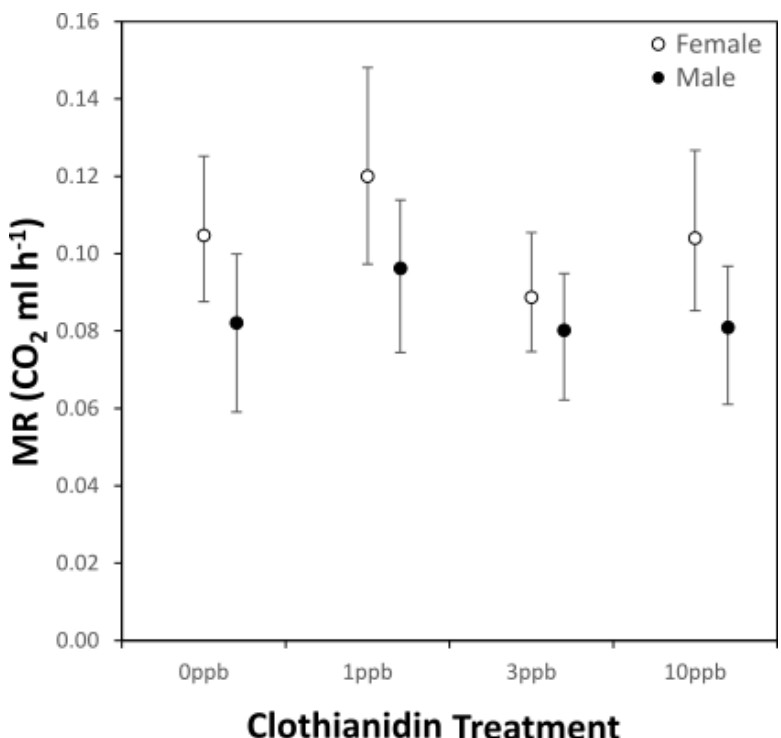

**Figure 4** **Metabolic rate for male (black circles, $n = 48$) and female bees (open circles, $n = 54$) exposed to varying concentrations of clothianidin during development.** Data are least-square means ($\pm$CI) estimated by an ANCOVA performed for each sex, with treatment and gas exchange pattern as a main effect and log-transformed adult body mass as a co-variate (Females = 72.03 mg, Males = 46.57 mg).

**Table 3** **Scaling relationship between wet body mass and metabolic rate for male and female bees engaged in continuous and discontinuous gas exchange.**

| Sex | Gas exchange | Log$_{10}$ slope | Log$_{10}$ intercept | Mass scaling relationship | Regression equation |
|---|---|---|---|---|---|
| Male | Continuous (CGE) | 1.78 | −0.77 | $MR = 0.170 \times M^{1.78}$ | $R^2 = 0.432, F_{1,6} = 3.804, p = 0.109$ |
| | Discontinuous (DGE) | 1.05 | −0.07 | $MR = 0.851 \times M^{1.05}$ | $R^2 = 0.278, F_{1,40} = 15.021, p < 0.001$ |
| Female | Continuous (CGE) | 1.24 | 0.03 | $MR = 1.072 \times M^{1.24}$ | $R^2 = 0.557, F_{1,16} = 18.837, p < 0.001$ |
| | Discontinuous (DGE) | 0.66 | 0.46 | $MR = 2.884 \times M^{0.66}$ | $R^2 = 0.136, F_{1,35} = 5.374, p = 0.027$ |

*versus* DGE (Fig. 3C, Table 3; ANCOVA, Breathing Pattern × Log-body mass, Females: $F_{1,49} = 2.054, p = 0.158$; Males: $F_{1,44} = 0.887, p = 0.351$), though bees engaged in CGE had a significantly higher metabolic rate (Table 3; ANCOVA, Breathing Pattern, Females: $F_{1,50} = 215.839, p < 0.001$; Males: $F_{1,45} = 72.212, p < 0.001$). For both sexes there was no significant difference in metabolic rate between clothiandin treatments (Fig. 4, ANCOVA, Treatment, Females: $F_{3,47} = 1.767, p = 0.166$; Males: $F_{3,42} = 0.819, p = 0.491$).

## DISCUSSION

Concerns have been raised regarding the impact that exposure to pesticides such as neonicotinoids have on bee health and non-target organisms more generally (*Pisa et al.,*
*2014*). Until recently, the majority of studies and regulatory tests have focussed on the effects of short term, acute exposure in adult bees, with a bias towards commercially reared social species such as honeybees and bumblebees. Given that neonicotinoid pesticides have been detected not only in the pollen and nectar of treated crops but also of wild flowers growing in the margins of fields, as well in ornamental garden plants (*Botías et al., 2015*; *Mogren & Lundgren, 2016*; *David et al., 2016*; *Long & Krupke, 2016*; *Lentola et al., 2017*) a host of bee species are likely to encounter these chemicals when foraging (*Hladik, Vandever & Smalling, 2016*; *Botías et al., 2017*), and yet we still know relatively little about the sensitivity of wild bees to neonicotinoids.

Here we examined the effect of chronic, larval exposure to neonicotinoids on the development and survival of a solitary bee, *Osmia bicornis,* under controlled laboratory conditions. We found that exposure to clothianidin at doses representative of the concentrations detected in field-collected pollen and nectar had no effect on development time or the efficiency with which larvae converted pollen into wet body mass. Overall, developmental mortality was similar between treatment groups (*ca.* 20%). A higher proportion of control group pupae failed to eclose, which elevated the overall mortality of this group above that of the treatment groups (*ca.* 30% in total). Once cocoons were placed back into a warmed incubator the following spring, there was no difference between treatment groups in the time to emerge, and adult body weight did not differ between exposed and non-exposed bees. Overall, our findings suggest that larvae of *O. bicornis* are insensitive to clothianidin exposure up to concentrations of 10 ppb.

A study of a North American bee of the same genus *(Osmia lignaria),* using a similar method of pollen contamination as our study, found that while the timing of discrete larval development stages was marginally affected by imidacloprid exposure, this effect was only apparent above 30 ppb. Indeed, even at 300 ppb, a concentration several orders of magnitude higher than that routinely detected in the field, no differences in survival to adulthood or adult body weight were observed (*Abbott et al., 2008*). In the same study, no effect of clothianidin exposure (3–300 ppb) on another species of megachilid bee, *Megachile rotunda,* was observed. *Abbott et al. (2008)* reared *O. lignaria* both in the laboratory and outdoors and no differences were observed in survival, though bees reared indoors did develop more quickly. Our study was conducted entirely in the laboratory to permit tight control over pesticide exposure and the rearing environment. Larvae were reared at a constant temperature of 20 °C, based on previous findings regarding optimal temperatures for normal *O. bicornis* development (*Radmacher & Strohm, 2011*). It remains to be tested whether fluctuating temperatures and varying humidity experienced under natural conditions would disproportionately affect clothianidin-exposed *O. bicornis* relative to non-exposed bees, though this seems unlikely given that development times did not differ from those reported for bees reared outdoors (*Raw, 1972*; *Radmacher & Strohm, 2011*).

Evidence from honeybees suggests that larval exposure to neonicotinoids can lead to physiological and behavioural changes in adult bees, though the mechanism(s) underpinning this are still not well understood (*Yang et al., 2012*; *Peng & Yang, 2016*). As a result, we were also interested in whether larval exposure to clothianidin leads to latent effects in adult bees. Because changes in the function of nervous tissue can affect

basic autonomic processes such as thermoregulation or respiration (*Sláma & Miller, 1987*; *Kestler, 1991*; *Belzunces, Tchamitchian & Brunet, 2012*; *Hatjina et al., 2013*; *Karise & Mänd, 2015*), we measured the metabolic rates of adult bees on the day of emergence. Relatively little is known about the effects of developmental pesticide exposure on adult insect metabolic rates, but acute exposure has been found to alter respiratory rhythms, particularly the propensity to engage in discontinuous gas exchange (*Zafeiridou & Theophilidis, 2004*; *Hatjina et al., 2013*; *Tomé et al., 2015*). We found that while the majority of bees engaged in discontinuous patterns of $CO_2$ release, a small proportion of bees breathed continuously throughout the observation period, but this propensity was not affected by clothianidin exposure. It also did not appear to reflect differences in activity within the recording chambers. Interestingly, female bees were almost three times more likely to engage in continuous gas exchange than males. The factors underlying switches between continuous and discontinuous patterns of gas exchange in insects are still widely debated (e.g., *Contreras & Bradley, 2009*), and so the reason why female bees were less likely to engage in discontinuous gas exchange is not clear, though could be related to sex-specific differences in body mass causing females to produce more $CO_2$ at rest (*Terblanche et al., 2008*).

The scaling of metabolic rate was identical across bees engaged in continuous *versus* discontinuous gas exchange, though metabolic rates were higher for bees exchanging $CO_2$ continuously. When metabolic rates were compared across treatments, we observed no significant treatment effect, though bees exposed to 3 ppb and 10 ppb clothianidin did have lower metabolic rates, on average. This suggests that the respiratory system of *Osmia bicornis* is unaffected by larval clothianidin exposure, though it remains to be tested whether other physiological or behavioural processes are affected. To our knowledge, the current study is the first to measure metabolic rates in *O. bicornis*. Previous studies have used flow-through respirometry to measure oxygen consumption at various stages throughout the life cycle of the cavity nesting bees, *O. lignaria* and *Megachile rotunda*, utilising the rate of oxygen consumption as a proxy for diapause intensity (*Kemp, Bosch & Dennis, 2004*; *Sgolastra et al., 2010*; *Sgolastra et al., 2011*; *Yocum et al., 2011*). Scaling relationships were not reported in these studies, therefore ours is the first to describe the intraspecific scaling of metabolic rate with body mass in a Megachilid bee, a measure which can be extremely useful in understanding the physiological functioning of an organism, as well as resistance to stress (*Burton et al., 2011*).

Our results contrast with previous observations of adverse effects of adult neonicotinoid exposure in this species (*Sandrock et al., 2014*; *Rundlöf et al., 2015*). Comparative studies have found adult *O. bicornis* to be more sensitive to clothianidin-spiked nectar than either honeybees (*Heard et al., 2017*) or honeybees and bumblebees (*Sgolastra et al., 2016*) though effects became apparent over different time-scales. A study of the closely related *O. cornuta* revealed that acute exposure to clothianidin can impair navigational behaviour under laboratory conditions (*Jin et al., 2015*). *Sandrock et al. (2014)* provided caged adult bees with sucrose solution spiked with thiamethoxam and clothainidin and found that females produced fewer nests, with fewer brood cells than those completed by female bees provided with uncontaminated sucrose solution. Larval and overwintering mortality was found to be higher in the offspring of neonicotinoid-exposed bees, such that fewer adult bees,

and proportionally more males, emerged the following spring. However, as in our study, offspring body size did not differ between control and neonicotinoid-exposed bees. Because pollen provisions in Sandrock et al.'s study were not found to contain neonicotinoids, the levels that larvae themselves were exposed to during development is not clear. Indeed, it may be that the reduction in offspring emergence is more attributable to effects on adult bee physiology or behaviour than to effects on the larvae. For example, consuming contaminated nectar may have impaired adults' provisioning ability (*Feltham, Park & Goulson, 2014*; *Gill & Raine, 2014*; *Stanley et al., 2015*), sex allocation (*Whitehorn et al., 2015*) or gamete viability (*Straub et al., 2016*),  as has been observed in other bee and wasp species.

Though there are clear methodological differences between previous studies of adult exposure to neonicotinoids and that presented here, both in terms of experimental design and the particular neonicotinoid and/or concentration used, our results tentatively suggest that the effects of neonicotinoid exposure may be less severe for *O. bicornis* larvae than for adults. Certainly in honeybees it has been suggested that larvae may be more tolerant to neonicotinoids than adult bees (*Yang et al., 2012*), which has been proposed to arise from differential nACh-receptor expression across developmental stages (*Thany et al., 2003*; *Thany & Gauthier, 2005*). Additionally, certain higher-level structures targeted by neonicotinoids, such as the mushroom bodies, are not fully developed in bee larvae (*Farris et al., 1999*). The latent effects of larval exposure to neonicotinoids on adult cognitive ability remain to be tested in *O. bicornis*.

Our findings contribute to an accumulating body of evidence showing that the impacts of neonicotinoid exposure can be both species specific and dependent on developmental stage, and serve as a caution against evaluating the toxicity of a particular pesticide based on the findings from a single 'model' species such as honeybees or bumblebees. In this study we considered the impact of a single pesticide in isolation, but in reality it is likely that bees are exposed to a suite of agrochemicals via the pollen and nectar they collect (*David et al., 2016*; *Botías et al., 2017*); with synergistic effects of combined exposure to neonicotinoids and fungicides having already been observed in adults of *O. bicornis* and the related *O. cornifrons* (*Biddinger et al., 2013*; *Sgolastra et al., 2016*). Therefore more data are needed to quantify the exposure risk of wild bees foraging and nesting in arable landscapes and the contribution of such stressors to ongoing bee population declines.

## ACKNOWLEDGEMENTS

The authors wish to thank Arthur David for screening pollen samples for neonicotinoid residues. We would also like to thank Professor Erhard Strohm and Chris Whittles for advice on rearing bees, Thomas David and Douglas Morrison for assistance with developmental observations, and two anonymous referees for their comments on an earlier version of the manuscript.

### Funding

This work was supported by the Biotechnology and Biological Research Council (No. BB/K014579/1) and the Association for the Study of Animal Behaviour. The funders had no role in study design, data collection and analysis, decision to publish, or preparation of the manuscript.

### Grant Disclosures

The following grant information was disclosed by the authors:
Biotechnology and Biological Research Council: No.BB/K014579/1.
Association for the Study of Animal Behaviour.

### Competing Interests

The authors declare there are no competing interests.

### Author Contributions

- Elizabeth Nicholls conceived and designed the experiments, performed the experiments, analyzed the data, wrote the paper, prepared figures and/or tables.
- Robert Fowler contributed reagents/materials/analysis tools, reviewed drafts of the paper.
- Jeremy E. Niven and James D. Gilbert conceived and designed the experiments, contributed reagents/materials/analysis tools, reviewed drafts of the paper.
- Dave Goulson conceived and designed the experiments, reviewed drafts of the paper.

### Data Availability

Figshare: https://figshare.com/articles/LICOR_Output/4965095.
DOI: http://dx.doi.org/10.6084/m9.figshare.4965095.

### Supplemental Information

Supplemental information for this article can be found online at http://dx.doi.org/10.7717/peerj.3417#supplemental-information.

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
