# Peer review of "Larval exposure to field-realistic concentrations of clothianidin has no effect on development rate, over-winter survival or adult metabolic rate in a solitary bee, Osmia bicornis"

_PeerJ, doi:10.7717/peerj.3417_

## Round 0.1 · original submission · Minor Revisions

Both reviewers raise some useful points for you to consider in your revisions. I would also add that you need to identify in the introduction why it is important to compare continuous and discontinuous gas exchange here. Also did you measure bee activity when assessing the respirometry rates? This will be useful particularly to know if bees that are more active are using continuous gas exchange, versus more sedentary ones. I also think that lines 327-332 do not belong in the results.

Line 236 - are you sure the chambers that your bees were kept in were 1.5ml?

Reviewer 1 ·

Basic reporting

A. It would be helpful to have the clothianidin concentrations stated in the abstract. This helps readers immediately assess relevance. If not, then I think it certainly belongs in the introduction (where concentrations found in OSR and field margins should also be stated).
B. The introduction provides an excellent lead into the study and a valuable update.

Experimental design

A_ The experimental design is good and well reported
B_ Line 186: Isn’t any sugar provided?
C_ Line 195: Please include what information is presented in Peters 2017.
D_ Lines 201-203: Is this correct? 10% acetone is very high – is it not toxic? It would be helpful to comment on the UT survival rates to normal survival rates.

Validity of the findings

The data are valid so long as caveated by the fact that some deficit may be hidden until adulthood and that this could not be investigated. This is particularly possible as the impact of neonicotinoids is largely on cognitive function and this has not been challenged in the current study.

Additional comments

A. Line 445: Some mention of potential effects on sexual bias or preferential loss of adults should be included as alterations in sex ratio have been observed following exposure to neonicotinoids in parasitic wasps (Whitehorn 2015), honey bees (Henry 2015) and bumble bees (Moffat 2016). Therefore, a potential impact could be delayed until breeding and sex provisioning, particularly if these require cognitive function (eg. Seen in N. vitripennis, Whitehorn 2015).
B. This is a very important study that builds our knowledge that not all insect species are the same with respect to vulnerability and each neonicotinoid is also different. This same pattern of uncertainty extends to all chemical groups and species. Confidence requires empirical knowledge, not extrapolation.

Reviewer 2 ·

Basic reporting

no comment

Experimental design

L244 provide a short description of the measurement procedure. Did you observe behaviour of the bees during metabolic measurement? Differentiation between active and resting behaviour would be helpful as gas exchange patterns depends also on behaviour (activity). Dicontinous gas exchange is mainly observed in resting insects and contious gas exchange often in active (e.g. honeybees).

Validity of the findings

L350 I would strongly suggest to analyse metabolic rate data depending on gas exchange patterns (CGE and DGE independent from each other, do not mix pattern!) and clothianidin treatments and present results at least in supplementary materials.

Additional comments

General comments to the authors

The authors represent an very interesting and important study concerning the discussion of impact of neonicotinoids on bee health. The experiments are well planned and conducted and reveal somewhat surprising findings. The manuscript is well suited for publication in PeerJ after a minor revision.

Special comments

Abstract
L44 Is there a significant difference in development time? If not I would say “no effect …”

Introduction
Overall well done, but a little bit to long and detailed. Try to shorten and to focus the aim of the presented study.

Materials and methods
L244 provide a short description of the measurement procedure. Did you observe behaviour of the bees during metabolic measurement? Differentiation between active and resting behaviour would be helpful as gas exchange patterns depends also on behaviour (activity). Dicontinous gas exchange is mainly observed in resting insects and contious gas exchange often in active (e.g. honeybees).

Results
L329 cite also Kovac et al. (2007) for gas exchange patterns in bees.
Kovac H, Stabentheiner A, Hetz SK, Petz M, Crailsheim K (2007) Respiration of resting honeybees.
J Insect Physiol 53:1250–1261
L350 I would strongly suggest to analyse metabolic rate data depending on gas exchange patterns (CGE and DGE independent from each other, do not mix pattern!) and clothianidin treatments and present results at least in supplementary materials.

Discussion
Overall well done
L404 consider activity of the bees!
L407 depends also on total amount of produced CO2


Table 1
provide explanation for abbrevations

---

## Round 0.2 · accepted · Accept

I am happy with the corrections made according to the reviewers comments. I think this manuscript should get a high level of interest. Congrats on an interesting piece of research.